# Molecular Mechanisms Underlying Twin-to-Twin Transfusion Syndrome

**DOI:** 10.3390/cells11203268

**Published:** 2022-10-17

**Authors:** Kazuhiro Kajiwara, Katsusuke Ozawa, Seiji Wada, Osamu Samura

**Affiliations:** 1Department of Obstetrics and Gynecology, The Jikei University School of Medicine, Tokyo 105-8471, Japan; 2Center for Maternal-Fetal, Neonatal and Reproductive Medicine, National Center for Child Health and Development, Tokyo 157-8535, Japan

**Keywords:** twin-to-twin transfusion syndrome, placenta, hypoxia, anemia, oxidative stress, ischemia-reperfusion injury, programmed cell death

## Abstract

Twin-to-twin transfusion syndrome is a unique disease and a serious complication occurring in 10–15% of monochorionic multiple pregnancies with various placental complications, including hypoxia, anemia, increased oxidative stress, and ischemia-reperfusion injury. Fetoscopic laser photocoagulation, a minimally invasive surgical procedure, seals the placental vascular anastomoses between twins and dramatically improves the survival rates in twin-to-twin transfusion syndrome. However, fetal demise still occurs, suggesting the presence of causes other than placental vascular anastomoses. Placental insufficiency is considered as the main cause of fetal demise in such cases; however, little is known about its underlying molecular mechanisms. Indeed, the further association of the pathogenic mechanisms involved in twin-to-twin transfusion syndrome placenta with several molecules and pathways, such as vascular endothelial growth factor and the renin–angiotensin system, makes it difficult to understand the underlying pathological conditions. Currently, there are no effective strategies focusing on these mechanisms in clinical practice. Certain types of cell death due to oxidative stress might be occurring in the placenta, and elucidation of the molecular mechanism underlying this cell death can help manage and prevent it. This review reports on the molecular mechanisms underlying the development of twin-to-twin transfusion syndrome for effective management and prevention of fetal demise after fetoscopic laser photocoagulation.

## 1. Introduction

Twin-to-twin transfusion syndrome (TTTS) is a rare complication that occurs in 10−15% of monochorionic multiple pregnancies, specifically in monochorionic (MC) twins. TTTS mainly occurs due to an imbalance in the supply of oxygen and nutrients across the vascular anastomoses of the placenta between the twins [1,2] (Figure 1). Occlusion of placental vascular anastomoses using fetoscopic laser photocoagulation (FLP) is considered effective for treating TTTS by directly coping with the underlying pathophysiology [3]. FLP is usually performed using a 2 mm fetoscope that is percutaneously inserted into the recipient sac under ultrasound guidance to coagulate the communicating vessels between the twins on the chorionic plate of the placenta to ablate all intertwin anastomoses [3]. Advances in FLP have helped achieve dramatically improved perinatal survival in TTTS [1,4,5,6,7,8]. The rate of survival of at least one twin and both twins at 6 months of age has been reported to be 93.2% and 71.6%, respectively [9]. However, the occurrence of fetal death after FLP remains a concern, and the incidence of neurodevelopmental impairment after FLP has been reported to be 11.1–13.3% in the systematic reviews of Rossi et al. and van Klink et al. [10,11]. Neurodevelopmental impairment is thought to occur due to hypoxia or ischemia-reperfusion injury (IRI)-derived oxidative stress. One of the primary etiologic factors responsible for donor death is placental insufficiency due to the lack of an adequate placental mass after FLP [12]. Clinically, fetal survival is difficult to predict because it can be affected by a combination of factors, including abnormal cord insertion, growth restriction due to differences in placental size, and preterm delivery due to postoperative rupture of the membrane [3,11]. Molecular mechanisms such as the renin–angiotensin system (RAS), vascular endothelial growth factor (VEGF)- and hypoxia-related pathways, and oxidative stress (OS) have been implicated in several placenta-associated diseases, such as twin anemia-polycythemia sequence (TAPS), selective fetal growth restriction (sFGR), and preeclampsia (PE). Applying these factors (such as VEGF, RAS, hypoxia-related pathways, and OS) to the pathogenesis of TTTS can better interpret the condition occurring in the TTTS placenta, as shown in Figure 2. Although little is known about the final outcomes, such as OS-induced cell death, resulting from these factors, elucidating the molecular mechanisms involved in the TTTS placenta may help facilitate novel preventive strategies for placental insufficiency, thereby leading to improved survival of the fetus. This review focuses on several reports on the molecular mechanisms related to TTTS compared with the pathogenesis of other placental complications, such as TAPS, sFGR, and PE.

## 2. Placental Expression of Vasoactive Proteins

The temporal and spatial expression of several growth factors, such as epidermal growth factor, insulin-like growth factor (IGF), transforming growth factor-β (TGF-β), placental growth factor (PlGF), and VEGF, control the development of the placental villous network [13,14]. VEGF has been characterized as an extremely potent proangiogenic factor acting on endothelial cells [13,15,16] and for this reason, numerous studies have demonstrated its effect on angiogenesis [17]. VEGF has been detected in villous cytotrophoblasts in the first trimester and in syncytiotrophoblasts throughout the remainder of pregnancy, indicating its involvement in placental development [18,19,20,21]. There is strong evidence that the overproduction of soluble fms-like tyrosine kinase 1 (sFlt1), a VEGF antagonist, is a major cause of the development of pathophysiological changes in the placenta of patients with PE [22,23,24]. It is well-known that in the two-stage placental model of PE, impaired remodeling of uterine spiral arteries induces poor placentation, with subsequent placental hypoplasia, that typically accompanies the abnormal uteroplacental circulation leading to a hypoxic placental condition [25,26]. As hypoxia stimulates the release of sFlt1, especially from trophoblasts but not from other types of cells such as human umbilical vein endothelial cells and villous fibroblasts, it has been reported that sFlt1 is a useful indicator of placental hypoxia [27,28,29]. Morine et al. investigated the concentration of VEGF in patients with TTTS treated with amnioreduction [14]. The study found that serum VEGF levels in the umbilical vein of TTTS-affected fetuses (both donors and recipients) tended to be higher than those in the unaffected control group, whereas there was no difference in the concentration of VEGF between donors and recipients in TTTS [14]. Kumazaki et al. reported that moderate to high levels of Flt-1 mRNA were detected only in the villi of the donor twin, and the expression of VEGF in the donor villi was greater than in the recipient. In contrast, the expression of PlGF mRNA and protein was minimal, probably due to the upregulation of angiogenesis in response to placental hypoperfusion [30]. The donor placenta can be in a state of hypoperfusion because of rapid blood transfer through the anastomotic vessels from the donor to the recipient in the TTTS placenta. This results in placental hypoxia (described in the “Hypoxia-related factors” section), thereby leading to the overproduction of sFlt1 [31,32,33]. It is still unclear whether placental hypoperfusion is also affected by defective spiral artery remodeling in the TTTS placenta (described in the “pathological changes” section) [34]. However, differences in VEGF expression in the donor and recipient placenta vary. Morine et al. and Kumazaki et al. reported that the expression of VEGF in the donor placenta was significantly higher than that in controls but was similar to that in the recipient [14,30], whereas Galea et al. reported that VEGF was significantly increased in the recipient compared to the donor [35]. It is well-known that hypoxia also induces VEGF upregulation [36,37]. Therefore, the expression of VEGF in donor and recipient placentas may vary among different studies because the interaction of blood volume change-mediated and hypoxia-mediated reactions in TTTS placentas may differ in each case. It is known that the sFlt1/PlGF ratio during the first trimester is increased in women who develop PE [38,39]. Additionally, since significantly higher levels of sFlt1 and sFlt1/PlGF ratio in twin pregnancies with an uneventful outcome were observed compared with singleton pregnancies, increased circulating levels of sFlt1 in a twin pregnancy may be due to increased placental mass or relative placental hypoxia as described later in the “Hypoxia-related factors” section [40,41]. When comparing maternal serum levels of sFlt1 and PlGF in MC twins with or without TTTS during the first trimester, the twins who later developed TTTS did not have increased levels of sFlt1 but had decreased PlGF levels when compared with the uncomplicated MC twins [42]. However, after TTTS developed, the second trimester maternal serum sFlt1 and the sFlt1 to PlGF ratio were significantly increased in the MC twin pregnancies with TTTS compared with the uncomplicated MC twin pregnancies Interestingly, decreased PlGF without increased sFLT1 like in TTTS was observed in the first trimester in patients who developed PE [38]. This suggests the presence of a placentation process that links TTTS and PE. In addition to sFlt1 and PlGF, plasma concentrations of soluble endoglin (sENG), which has been reported to be a factor involved in the pathogenesis of PE, was also higher in patients with TTTS than in non-TTTS [43]. Although the placental expression level of ENG in placentas with TTTS was similar to that of non-TTTS, ENG levels were higher in donor placentas than in recipient placentas [44]. Higher levels of sFlt1 and sENG, and lower levels of PlGF, in TTTS suggest that TTTS is an antiangiogenic state similar to PE [45]. However, it remains unknown whether the development of TTTS is caused by an abnormal angiogenic profile, or whether an antiangiogenic state is a consequence of TTTS. As sFlt1 significantly decreased several weeks after FLP for TTTS [33], the antiangiogenic profile seems to be a consequence of TTTS. Further studies are, therefore, needed to characterize the upstream pathogenic events leading to this angiogenic dysfunction, and they are currently a source of ongoing research.

## 3. Renin–Angiotensin and Endothelin Systems

There is a significant amount of literature indicating that discordant renin–angiotensin system (RAS) activation in TTTS is pivotal in its pathophysiology, as the RAS plays an important role in the regulation of arterial blood pressure and sodium and water homeostasis [46]. Paradoxical RAS activation has been reported in some studies, as noted below. RAS upregulation was observed in the donor’s kidneys in TTTS, presumably secondary to hypovolemia, and downregulation was reported in the recipient’s renal system due to hypervolemia [46,47]. Although the results of renin regulation in the kidney reflect the pathophysiology in TTTS, there is some confusion caused by the findings of renin plasma levels in donor and recipient. There are reports that the plasma renin level tended to be higher in recipients compared with donors, but this difference was not significant, and the renin concentration in both donor and recipient was higher than in controls (unaffected MC twins) [35]. Indeed, it is controversial whether renin is transferred from the donor to the recipient via placental anastomoses. The amount of renin transferred would be insufficient to result in high recipient levels of donor origin because renin has a short half-life and chronic transfusion flow is too low (3% of fetal cardiac output per shared cotyledon) [35,47,48], suggesting that the recipient is probably exposed to another source of RAS. The placentas of patients with TTTS showed higher expression levels of renin in the recipient than in the donor or controls on immunohistochemistry and Western blot [35]. These results indicate that the placenta may be the source of the high plasma renin levels in the recipient. In contrast, maternal plasma renin levels, which were slightly above the upper normal nonpregnant levels in TTTS, decreased after treatment with FLP [49], indicating that sufficient amounts of renin are secreted from the placenta to increase or decrease fetal blood levels. A microarray analysis of the TTTS placenta demonstrated that angiotensin-converting enzyme (ACE)1, a potent vasopressor, and ACE2, a potent counter-regulator to ACE1, were upregulated in the recipient compared with the donor [35,50,51,52]. However, it is not known which action, ACE1-mediated vasoconstriction or ACE2-mediated vasodilation, is predominant in the recipient placenta. The uteroplacental unit possesses not only renin and ACE but also all components of the RAS necessary for the generation of Ang I and II [53]. For example, Ang II type 1 (AT-1) receptors are localized in the cytotrophoblast and syncytiotrophoblast of the placental villi [54]. In an analysis of the TTTS placenta, microarray data demonstrated greater expression of Ang, AT-1, and Ang II type 1 (AT-2) receptors in the recipient compared to the donor, indicating activation of RAS in the recipient sector [35]. Taken together, these studies show that it is difficult to explain the paradoxical changes of renin, including higher plasma renin concentration in the donor and recipient, downregulation in the recipient kidney and donor placenta, and upregulation in the donor kidney and recipient placenta, only by the presence of fetal blood volume discordance caused by anastomotic vessels. This indicates that another pathway or pathways must be involved.

The endothelin-1 (ET-1) system is one of the most potent vasopressor mechanisms and is as well-known as RAS [55]. ET-1 was reported to have not just a vasoconstrictive effect but also multiple other effects on the RAS including a dose-dependent inhibition of renin synthesis, which directly stimulates aldosterone production [56,57]. In a fetus with TTTS, higher levels of ET-1 were also observed in the recipient’s amniotic fluid and cord blood than in the donor’s own samples collected by fetal blood sampling during pregnancy without treatment with FLP [32,58,59]. The ET-1 level in the donor was similar to that in non-TTTS twins [58,59]. It is unclear whether a higher ET-1 level in the recipient worsens the recipient’s systemic hypertension or might contribute to the recipient’s renal RAS shutdown [59,60]. Transplacental passage of ET-1 from the maternal circulation seems unlikely because ET-1 concentrations in the fetus have been found to be higher than in the mother [59]. Although the cord arterio–venous gradient suggests that higher concentrations are due to increased production by the placenta [59], a comparison of placental ET-1 expression showed no difference between donor and recipient [35]. This indicates other organ involvement, such as the fetal heart, which may produce ET-1 [61]. Interestingly, there was no difference in ET-1 levels in cord blood between donors and recipients after treatment with FLP [62]. The authors assumed that initial differences in ET-1 levels before FLP may have gradually diminished and disappeared after FLP. Because hypoxia has been reported to induce ET-1 [63], it is unclear if serum ET-1 levels were increased in recipients rather than donors simply due to hypoxia. Interestingly, it has been reported that cardiac hypertrophy and ET-1 upregulation were induced by mild hypoxia, indicating the presence of mild hypoxia in the recipient [64]. In summary, it is not known whether RAS and ET-1 activation of vasoconstriction of the fetal capillaries in the terminal villi is a regulatory mechanism of blood flow from the placenta to the recipient. Further research is required to confirm this concept.

## 4. Differences in RAS between PE and TTTS

Because transgenic mice expressing both human angiotensinogen and human renin induce a PE-like syndrome, the RAS has received considerable attention regarding the pathogenesis of PE, whose underlying causes are abnormal placental hypoplasia with hypoperfusion [65]. In humans, an analysis of the uterine placental bed demonstrated that Ang II, renin, and ACE mRNA expression were significantly enhanced in the preeclamptic uterine placental bed compared with normotensive placental beds. However, when comparing the expression in the placenta, there was no difference in RAS-related genes between PE and control placentas [66]. Reports on the maternal plasma renin activity in patients with PE have not been consistent [66], suggesting an insignificant role for the RAS in the placenta after PE has developed. Systemic symptoms in early-onset PE are believed to be primarily caused by an elaboration of proteins or factors by the placental tissue that enter the maternal circulation. These substances include sFlt1 and sENG as well as other RAS-related factors [67,68]. Altogether, in a placenta from PE where hypoxia is presumed, it appears that the RAS is increased during the developmental stage to cause its pathology, whereas in TTTS, the RAS appears to be increased as a result of the pathology. An in vivo experiment conducted by Lumbers et al. suggested the relationship between asphyxia induced by umbilical cord occlusion and renin activation; however, the underlying pathology of this model is IRI, suggesting a relationship between IRI and the RAS [69]. Thus, it seems that hypoxia alone does not activate renin in the placenta, but changes in blood volume, which are at the root of the pathogenesis of TTTS, seem to affect the RAS to a much greater degree.

## 5. Expression of ACE2 in TTTS and TAPS Placentas

Since ACE2 serves as a molecule for severe acute respiratory syndrome coronavirus 2 cell entry, numerous studies of ACE2 have been reported recently due to the SARS-CoV-2 pandemic [70]. ACE2 plays a key role in catalyzing the change of vasoconstrictor peptide ANG II to Ang-(1–7), thereby acting as a protective component of the RAS in opposing the overactivity of ANG II [71]. The involvement of ACE2 in placentation and its dysregulation has been implicated in pregnancy complications such as miscarriage, FGR, and PE [72]. ACE2 is also a useful factor to understand placental physiology as it is related to hypoxic conditions, and there is data on it from TTTS and TAPS. It has been reported that ACE2 was upregulated by hypoxia in several types of cells as well as in the human placenta [73]. In contrast, maternal hypoxia has been reported to induce decreased placental ACE2 mRNA in a mouse model [74]. Hypoxia inducible factor 1α (HIF-1α), which is upregulated by hypoxia, exerts its inhibitory effect on ACE2 through microRNA let-7b [75]. Activation of the HIF-1α signaling pathway under mildly hypoxic conditions would decrease ACE2 expression [76,77]. Interestingly, there is a difference in ACE2 expression in the placentas of TTTS and TAPS. Recently, Mao et al. analyzed the expression of ACE2 in the placentas of patients with TAPS excluding cases following FLP. They reported that ACE2 protein levels, which were expressed in villous trophoblastic and stromal cells, were consistently higher in anemic sections of the placenta compared to the corresponding polycythemic placental sections [78,79]. The authors concluded that increased ACE2 expression was induced by hypoxia based on previous data regarding autophagy activity in anemic sections [80] (described in the “Autophagic activity” section). This was opposite to the result from TTTS in the study by Galea et al. because the ACE2 expression was higher in the recipient placenta than in the donor, which in turn was higher than in a control MC twin [35]. Taken together, these results show that upregulation of ACE2 expression, as an adaptive stress response to hypoxia, can be associated with TAPS and TTTS. Additionally, while TAPS would be expected to have no increase or decrease in fetal blood flow, a donor fetus with TTTS would clearly have intervascular volume loss, which could have a stronger effect on the RAS in the placenta. In summary, hypoxia-mediated ACE2 upregulation may have occurred in both donor and recipient, or there may have been stimulation of the RAS by means other than hypoxia.

## 6. Is There an Anemic Condition in the Donor Placenta from TTTS?

Fetal hemoglobin Bart’s disease is a congenital disorder of red blood cells characterized by the presence of hydrops fetalis [81]. Tongprasert et al. reported that in cases of Bart’s disease, sFlt1 was not significantly different between affected and unaffected cases [81]. Maternal serum sFlt1 was significantly increased in pregnancies with mirror syndrome associated with fetal hydrops caused by rhesus isoimmunization, fetal cytomegalovirus infection, and cardiac failure [82]. In those reports, PE-related factors other than sFlt1 were also noted, including a higher level of sENG, a lower level of PlGF, and strong staining with antibodies against sVEGFR-1 in the syncytiotrophoblast, particularly [83]. The elevation of sFlt1 in mirror syndrome may be due to the placental hypoxia caused by villous stromal edema. Hydrops is frequently associated with villous edema [82,84,85,86], and it is characterized by high intracellular water content and high total placental water [87]. This results in a compression of the villous blood vessels by edematous villi or a thicker interface which leads to a reduction of the intervillous space and the intervillous blood supply with a subsequent reduction in the fetal oxygen supply and gas exchange [88,89,90]. It has been reported that the higher the severity of villous edema, the lower the umbilical artery cord pH would be [90]. Taken together, it seems that fetal anemia does not always result in villous stromal edema, but in severe cases such as hydrops, placental hypoxia, due to villous stromal edema, induces an increase in the level of maternal serum sFlt1. Although it is unclear whether central findings of edematous villi can be caused by the effect of sFlt1 on the placenta or by increased fetal systemic vascular resistance in fetal hydrops resulting in increased venous pressure, it appears clear that placental hypoxia due to the latter hypothesis increases maternal serum sFlt1.

In a report on TAPS, villi from the anemic twin placenta were slightly edematous, supporting the hypothesis that fetal anemia induces edema in the villi leading to hypoxia in the fetus [91]. However, villous stromal edema does not appear to be common in the placenta of TTTS. Because a preoperative elevated MCA-PSV was only present in 4.2% of donors and 3.2% of recipients in patients with TTTS, fetal anemia seems to be uncommon in the donors of TTTS [92]. It is unclear why the donor fetus is not anemic (high MCA-PSV velocity) in TTTS. It is assumed to be purely due to the reduced blood supply from the placenta, which, combined with the RAS activation in the donor, results in an overall hypodynamic state with relative hemoconcentration. This reflects the status as nonanemic (only a slight decrease in hemoglobin level), whereas fetal blood volume does not change in TAPS. The study noting that the hematocrit level of the donor was in the normal range despite a slight decrease in the hemoglobin level [93,94]. In the TAPS placenta, absence of RAS upregulation in the donor allows the vessels to maintain a colloidal osmotic pressure and intravascular volume that results in the decreased level of hemoglobin and hematocrit [95]. This means that the degree of hemoconcentration is stronger in TTTS due to the loss of intravascular volume through the anastomotic vessels on the placenta. In another possibility, it has been suggested that pericytes control the distribution of red blood cells at capillary bifurcations and constantly adapt the local capillary diameter based on local cellular needs [96]. Although the fetal capillary diameter of the donor and anemic twin sectors was smaller compared to the recipient and polycythemic sectors [91,97], compensatory downregulation of the RAS in the hypoxic donor placenta might maintain the diameter of the fetal capillaries in the donor placenta. Unfortunately, there is no data to support this hypothesis. Further examination of the relationship between fetal capillary diameter and RAS-related factors in TTTS and TAPS is, therefore, required.

## 7. Pathological Changes

There is little evidence about the macro and microanatomy of the underlying shared cotyledons, which mediate interfetal transfusion in TTTS [97]. Wee et al. reported the histomorphometry of terminal villi in shared and nonshared cotyledons in TTTS [97]. The terminal villi from the donor twin’s placental sector demonstrated a reduced average terminal villus diameter, smaller capillaries, a smaller degree of vascularization, and a larger fetomaternal diffusion distance compared to the recipient’s sector. Kumazaki et al. reported that increased syncytiotrophoblastic knots, shrinkage of villi, increased perivillous fibrin deposition, villous infarction, and villous hypercapillarization were observed in the villi of the donor [30]. These findings were also detected in the terminal villi from a growth-restricted fetus without TTTS compared to the appropriate for gestational age (AGA) twin’s territory. In comparison, shared and nonshared cotyledons are histomorphometrically similar in uncomplicated MC twin pregnancies, indicating that the shared cotyledon supplied by an arterio–venous anastomosis (AVA) in a non-TTTS placenta is simply a normal cotyledon [97]. In this study, birthweight discordance and a discordance in the size of the placentas were not related to the difference in terminal villus diameter across the donor and the recipient placental sectors (delta (Δ)diameter). The Δdiameter was correlated significantly with the number of AVAs from the donor to recipient minus the number of AVAs from the recipient to the donor (ΔAVA) in TTTS. That the Δdiameter was influenced by placental sector discordance and birthweight discordance in sFGR suggests that the differences observed in TTTS are due to placental vascular anatomy, or more precisely, to interfetal transfusion, as opposed to birthweight discordance and placental sector size. The authors suggested that these terminal villi findings could occur due to altered local intraplacental hemodynamics rather than developmental abnormalities such as an abnormal uteroplacental circulation. In contrast, Matievic et al. measured the resistance index (RI) of the spinal artery by ultrasound and reported that the RI of the spiral artery was higher in the donor than the recipient cotwin. This suggested an impaired endovascular trophoblast invasion of the myometrial component of the maternal spiral arteries [34]. However, the spiral artery RI in the FGR twin of non-TTTS was also higher than that of AGA cotwins, suggesting that the difference is not specific to TTTS. Although there was no data on the pathological analysis of the placenta in this report, the author suggested that a reduced blood supply from the spinal artery in the donor placental sector may affect the development of the smaller placenta compared to that of the recipient. Fetal thrombotic vasculopathy (FTV) was reported to be implicated in other perinatal and neonatal complications, including abnormal fetal heart rate, stillbirth, birth asphyxia, and neonatal coagulopathy [98,99,100]. An increased number of placental FTV cases have been reported in twin gestations whether complicated with a clinical diagnosis of TTTS or not [101]. Chan et al. analyzed placentas to identify the incidence of fetal vessel thrombosis in monochorionic twins with fetal vascular anastomoses and TTTS [102]. Fetal vessel thrombosis, defined as avascular chorionic villi with fibrin thrombi in the muscular vessels of the chorionic plate or stem villi, was strongly associated with sFGR in MC twins, but the same association was not observed in dichorionic twins. However, fetal vessel thrombosis showed no relationship with either TTTS or vascular anastomoses. Moreover, it should be noted that the Quintero staging system, which is the most commonly used classification system for TTTS [103], was not used in this study. Because velamentous cord insertion is known to be associated with fetal vessel thrombosis in twin pregnancies [104], FTV sometimes occurs in placentas in TTTS, but it may be more associated with FGR-related conditions, such as umbilical cord occlusion and a hypercoagulable state such as antiphospholipid antibody syndrome [102]. Villous immaturity in the donor placenta of TTTS has also been reported [105]. Delayed villous maturation (DVM) was also reported in acardiac twinning which presents the extreme end of the spectrum of TTTS [106]. Villous tree maturation develops by the approximation of syncytiotrophoblasts with the villous capillary endothelium and constitutes the most efficient sites for gas exchange in the placenta. Therefore, delayed maturation of the villous tree can lead to deficient vasculosyncytial membranes which has been associated with a high incidence of hypoxic complications [107]. CD-15 immunohistochemistry has been used to detect DVM. Further analysis of the TTTS placenta using CD-15 may be useful to determine whether placental hypoxia was derived from DVM in the donor placenta [107].

## 8. Hypoxia-Related Factors

The assessment of fetal tissue venous oxygenation (StO2%) in utero during FLP for TTTS via visible light spectroscopy revealed that the StO2% of the placental surface was significantly lower in the donor than in the recipient twin [108]. In a report on the analysis of twin placentas, the expression of HIF-1α and HIF-2α, which are thought to be excellent markers for tissue hypoxia, was not different between the placentas from twin or singleton pregnancies [41,109,110]. The authors suggested that the increased circulating level of sFlt1, which was twice the level in twin pregnancies as in singleton pregnancies, without upregulation of HIF-1α, indicated that the placentas from uncomplicated twin pregnancies are generally not hypoxic, and that each trophoblast is not programmed to produce more of the antiangiogenic protein sFlt1 [41]. In other words, oxygenation of each placenta is normal as it consists of the production of sFlt1 by each trophoblast; however, the overall sFlt1 level might be elevated because there is twice as much placenta in a twin pregnancy. In contrast, sFlt1 was found to be higher in TTTS than in non-TTTS [43], and the level significantly decreased several weeks after FLP, indicating the presence of hypoxia in the TTTS placenta [33]. In the analysis of other hypoxia-related factors in twin pregnancies, HIF1α expression was elevated in the placental sector of the smaller fetus in sFGR pregnancies [111,112,113], indicating the presence of more severe hypoxia than in an uncomplicated twin pregnancy. The results of aberrant hydroxymethylation of angiopoietin-like 4, a hypoxia-responsive gene regulated by HIF-1α, and increased expression of miR-199a, a master regulator of a hypoxia-triggered pathway targeting HIF-1α in the smaller placental sectors of sFGR, are consistent with the results noted above [114,115]. There is only one report noting evidence that HIF-1α was primarily observed in trophoblastic cells and villus capillary endothelial cells in the donor placenta from TTTS compared with that in the control placenta from non-TTTS twin pregnancies, but the expression of HIF-1α in the recipients tended to be similar to the controls [116]. It is unknown whether the recipient sector experienced hypoxia, but it is possible that mild hypoxia may be present in the recipient placenta due to high blood viscosity, hydrostatic–osmotic, and amniotic pressures [117]. These may all contribute toward placental bed hypoxia since the presence of hypoxia in the recipient is a potent stimulus for the production of ET-1 and ACE2 (described in “Renin–angiotensin and endothelin systems” and “Expression of ACE2 in TTTS and TAPS placenta” sections) [35,118,119]. Further studies comparing other tissue hypoxic markers between normal placentas and the placentas of recipients are necessary to confirm this hypothesis. In addition to the phenomenon that HIF also reacts with other regulatory pathways, such as insulin/IGF and glucose transporters, trophoblasts and mesenchymal stem cells may respond differently to hypoxic conditions even within the same placenta. This may be the cause of conflicting results when assessing specific protein expression as a tissue [113,120].

## 9. OS

It is well-known that pregnancy increases OS, a phenomenon primarily produced by a normal systemic inflammatory response, which results in high levels of circulating reactive oxygen species (ROS). These play an important role as secondary messengers in many intracellular signaling cascades [121]. The placenta is not only a major source of ROS during pregnancy, but also regulates its production [122]. Usually, the OS caused by mitochondria in the placenta increases as gestation advances, and it is neutralized by antioxidants. However, when the OS surpasses the antioxidant defense in the placenta, oxidative damage could propagate to distal tissues. There is an imbalance in the OS in some pregnancy complications such as FGR, PE, and diabetes [123]. Several reports have noted that twin pregnancies may be in a state of increased OS. Gür et al. reported that increased levels of malondialdehyde (MDA), a marker for lipid peroxidation which is usually generated by ROS, and decreased antioxidant ability of vitamins A and E were observed in ewes with twin pregnancies compared with those with singleton pregnancies [124]. The same result was noted in humans with an increase in MDA levels in twin pregnancies compared with singletons [125]. In sFGR MC twin pregnancies, the indicators of OS such as MDA, 8-hydroxydeoxyguanosine (8-OHdG), and COX2 were elevated in the placental sector of the growth-restricted fetus [111,126]. OS may affect the replication of mitochondrial DNA (mtDNA) which has been shown to be increased in the placenta from patients with FGR [127]. In a study of twin pregnancies, it has been reported that fetal blood from the sFGR twin contains significantly higher mtDNA [128], indicating a higher state of stress in the sFGR fetus compared to a singleton fetus. Similarly, mitochondrial damage was observed in the smaller twin in sFGR without TTTS, confirmed by examination of the mtDNA *MTND1* copy numbers and electron microscopy of the trophoblast cells [126]. Chang et al. examined the placental mtDNA fold changes (FC) between smaller and larger twins using real-time quantitative PCR and reported that the placental mtDNA FC were significantly higher in TTTS twins with sFGR compared to that of MC twins without TTTS but with sFGR [129]. This result indicates that hypoxia in TTTS is more severe than in sFGR. The mtDNA FC in the placental sector of the sFGR (donor) twin with TTTS were 1.57-fold higher than that of the larger (recipient) twin [129]. However, the study found no difference in the FC between the donor and recipient in cases of TTTS without sFGR. This is probably because OS is improved by FLP in TTTS with the same results as for just a sFGR placenta. Heme oxygenase 1 (HMOX1), which is induced by the nuclear factor erythroid 2 like 2 (NFE2L2), is a major antioxidant that protects cells from OS. It has been reported that NFE2L2 or HMOX1 deficiency is related to gestational complications including PE and FGR [130,131,132,133]. As well as NFE2L2 being upregulated by hydrogen peroxide-induced OS in HTR-8/SVneo trophoblast cells [134], NFE2L2 and HMOX1 were both upregulated in the placental sectors of the smaller fetus in sFGR pregnancies [112]. In the case of TTTS, a severe OS condition is assumed to be present in the donor placenta, because some OS-induced changes were observed in the placenta of the smaller twin of sFGR (Figure 3). Therefore, further research is needed to determine and confirm whether the antioxidant system is reduced in TTTS donor placentas as it is in PE and sFGR placentas [135].

## 10. Autophagic Activity

Specific programmed cell death is induced in response to OS occurring within cells. Autophagy, which is a predominantly cytoprotective process, has not only been linked to programmed cell death but also is upregulated under stress conditions, including cell starvation, hypoxia, and OS [136]. Moreover, autophagy plays multiple key roles in placental development [137]. It has been reported that microtubule-associated protein 1A/1B-light chain 3 (LC3) proteins were observed in the human placenta [138,139]. LC3-II, the conjugated form of LC3, is a standard marker for the autophagosome and can reflect autophagic activity. Chang et al. reported that autophagic activity was increased in the placental sector of sFGR fetuses, especially in an sFGR twin with abnormal umbilical artery Doppler flow [140]. Similarly, LC3II was increased in anemic placental sectors compared with those in polycythemic sectors in TAPS placentas [80]. These differences may result from relative hypoxia in the placental sector of the anemic fetus compared to the cotwin, because the expression of carbonic anhydrous (CA) IX protein, a hypoxia marker, was significantly higher in the anemic sectors of a TAPS placenta than in corresponding polycythemic sectors [80]. However, in the placenta from patients with TTTS, after eliminating intertwin vascular anastomoses by FLP, discordant placental autophagic activity was not present when comparing AGA and sFGR fetuses [141]. This suggests that FLP can improve placental stress, especially in the donor placenta, and lead to a decrease in the intertwin autophagic activity discrepancy. Although there are only several reports noting increased placental apoptosis in the smaller fetus of discordant twins [142,143], an examination in TTTS was lacking. Which type of cell death is caused by excessive OS depends on the sensitivity to death of each type of cell. A previous report noted that ferroptosis, a recently discovered programmed cell death, was easily induced in trophoblasts compared to other types of cells [144]. Since OS usually causes some form of cell death, clarification of the process could establish a novel preventive strategy focusing on the inhibition of specific cell death. Therefore, further research on cell death in TTTS placentas is warranted.

## 11. IRI

It has been suggested that IRI in the placenta induces overproduction of oxidative agents [145,146] which may result in damage to lipids, since they are the first targets of oxidative damage [147]. Some pregnancy complications, such as FGR, could result from IRI-induced pathological changes [148,149]. There are reports of TTTS in which the antenatal demise of the recipient twin was followed by the rapid development of hydrops in the donor twin [150,151]. An in vivo experiment showing that hydrops was induced after the release of complete umbilical cord occlusion strongly suggests a relationship between IRI and hydrops [69]. Because reperfusion after tissue ischemia can induce the disruption of endothelial integrity with loss of fluid to the endothelial and interstitial spaces resulting in endothelial cell swelling and edema [152,153], subsequent development of hydrops fetalis in the surviving donor twin was likely due to IRI of the previously poorly perfused twin. Since it is difficult to determine whether IRI is truly present in TTTS, it is quite important to focus on IRI-mediated tissue damage because IRI can be related to neurodevelopment impairment in TTTS. This was described in the study of Palanisamy et al. who noted that neurological damage could be caused by transient hypoxia-ischemia in a rat pregnancy model [154]. Furthermore, evidence that hypoxia/reoxygenation during pregnancy increases ferroptosis in the placenta in vivo strongly suggests the presence of IRI-mediated OS in the placenta [155].

## 12. Conclusions

In summary, the possible mechanisms of this review are shown in Figure 2. They suggest that hypoxia is definitely present in the donor placenta of TTTS, but intervascular volume loss and consequent activation of the RAS leads to a hemoconcentrated state, making anemia unlikely. Instead, the presence of OS due to hypoxia is suggested, but it is still unclear which type of cell death is induced. Figure 4 highlights the expression profiles in TTTS that were reviewed in this study. Since multiple pathways such as VEGF, RAS, and HIF are involved in the pathogenesis of TTTS, it is difficult to easily explain the physiological changes of this unique disease. Activation of these factors and tissue responses may ultimately result in OS and cell death in trophoblast cells, but there are few reports on the impact of cell death, especially in trophoblasts in TTTS. Therefore, further research on the evaluation of cell death is needed to develop an effective management strategy to reduce fetal mortality in TTTS.

## Figures and Tables

**Figure 1 cells-11-03268-f001:**
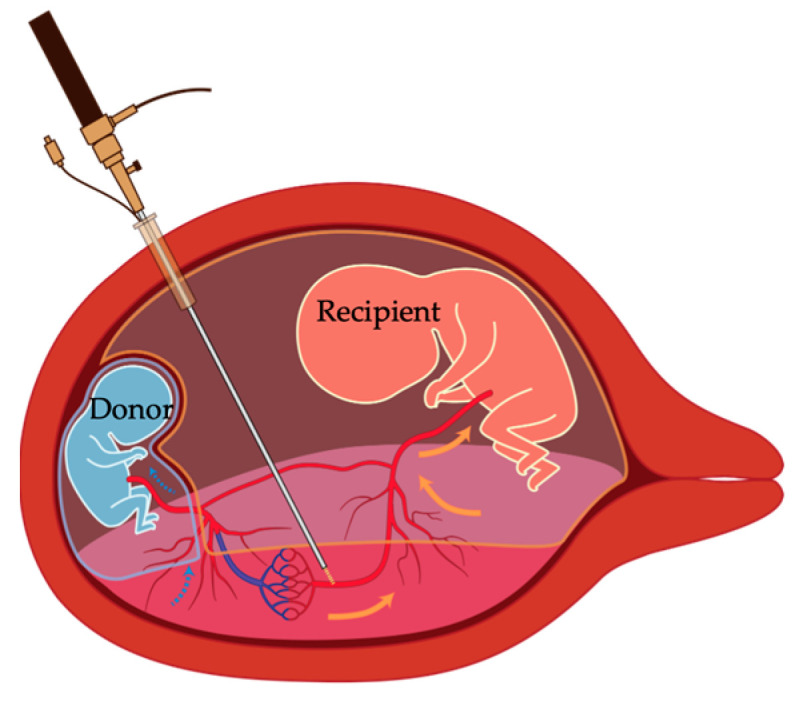
Pathology of TTTS. Circulatory imbalance is caused by a net transfusion of blood from donor to recipient in the presence of deep vascular anastomoses and develops chronically between hemodynamically connected monochorionic twin fetuses.

**Figure 2 cells-11-03268-f002:**
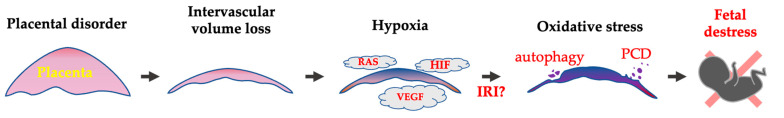
Events assumed to be occurring in the donor placenta. Abbreviations: HIF: hypoxia inducible factor; IRI, ischemia-reperfusion injury; PCD, programmed cell death; RAS, renin–angiotensin system; VEGF, vascular endothelial growth factor.

**Figure 3 cells-11-03268-f003:**
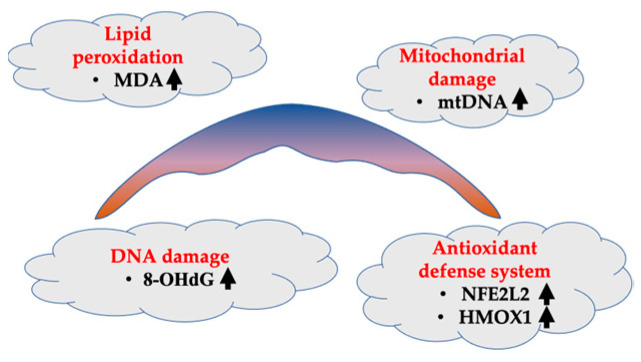
OS-induced changes in the placenta of a smaller twin in sFGR. Abbreviations: HMOX1, heme oxygenase 1; MDA, malondialdehyde; mtDNA, mitochondrial DNA; NFE2L2, nuclear factor erythroid 2 like 2; 8-OHdG, 8-hydroxydeoxyguanosine.

**Figure 4 cells-11-03268-f004:**
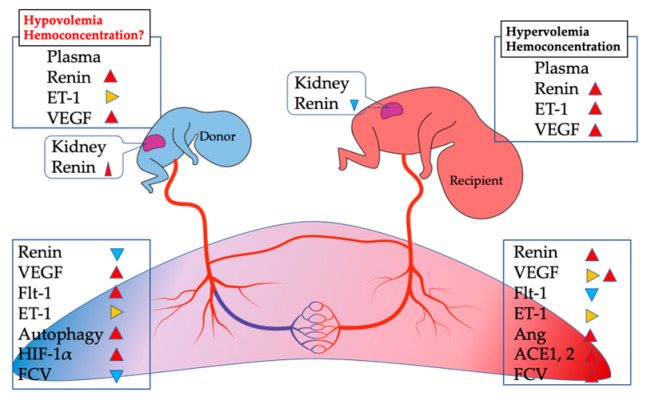
A schematic depicting the expression profile of TTTS-related factors in donor and recipient. Red, blue, and yellow triangles indicate high, low, and equal expression or concentration, respectively. Abbreviations: ACE, angiotensin-converting enzyme; Ang, angiotensin; ET-1, endothelin-1; FCD, fetal capillary diameter; Flt1, fms-like tyrosine kinase; HIF-1α, hypoxia inducible factor 1α; VEGF, vascular endothelial growth factor.

## Data Availability

Data is contained within the article.

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
