# Peer review of "Molecular Mechanisms Underlying Twin-to-Twin Transfusion Syndrome"

_cells, 2022, doi:10.3390/cells11203268_

Round 1
Reviewer 1 Report
This is a comprehensive review article around the molecular changes that occur in TTTS syndrome placentas. The summary of the literature is superb and very detailed, which will provide a great foundation and reference to the readership.
Two small suggestions I would like to make, please could the authors include a diagram early on in the article that depicts the pathology of TTTS so that the readership outside the field can follow. Secondly, please could the authors make Figure 2 larger and, if needed, add more detail even. The text is very dense and the figure will make it a lot easier to grasp the significant changes at a quick glance.
There are one or a few instances where the font changes all of the sudden, and the text does not read as smoothly. Please homogenize contributions by different co-authors to make the text more coherent in these few places.
Author Response
- Comment: Two small suggestions I would like to make, please could the authors include a diagram early on in the article that depicts the pathology of TTTS so that the readership outside the field can follow. Secondly, please could the authors make Figure 2 larger and, if needed, add more detail even. The text is very dense and the figure will make it a lot easier to grasp the significant changes at a quick glance.
Response: We thank the reviewer for the useful comment. Accordingly, we have performed added the figure that shows the pathology of TTTS. Additionally, we have also made the figure2 larger.
Inserted word (Page1, line4): (Figure 1)
Inserted figure (Page 2, line 4): Figure1
Inserted sentences in Figure 1 legend (Page 2, lines 20-23): Figure 1. Pathology of TTTS. Circulatory imbalance is caused by a net transfusion of blood from donor to recipient in the presence of deep vascular anastomoses and develops chronically between hemodynamically connected monochorionic twin fetuses.
2.Comment: There are one or a few instances where the font changes all of the sudden, and the text does not read as smoothly. Please homogenize contributions by different co-authors to make the text more coherent in these few places.
Response: We have changed the font of the main text and figures and figure legends.
Reviewer 2 Report
Dear Authors
I have read with interest the review about pathogenetic mechanism for monochorionic placental complications: it sounds well structured , complete and appropriate.
Author Response
- Comment: I have read with interest the review about pathogenetic mechanism for monochorionic placental complications: it sounds well structured , complete and appropriate.
Response: We thank the reviewer for the favorable comments. I believe that this manuscript will attract general and specialist readers interested in the potential medical applications of fetal therapy, as well as scientists researching placental biology.
Reviewer 3 Report
The authors have provided an extensive and comprehensive review of the available literature on the molecular mechanisms underlying twin twin transfusion syndrome. It is well organized and it reviews the available literature under multiple pertinent subheadings. The paper is certainly helpful in putting this complex disease process into better focus. As a reviewer I only have some simple suggestions for the authors.
In the abstract and introduction it is important to emphasize that you are highlighting the placental pathology and potential reasons for excess fetal loss. As a clinician who frequently treats TTTS with laser, I see the complications up close. Pre- and peri-viable delivery results in a significant proportion of the morbidity and mortality after laser as does placental insufficiency for the donor fetus as you do point out. Another area that is present is the significant cardiomyopathy seen in the recipient twins. So my point is the excess fetal loss despite laser comes from multiple sources rather than confined to just factors within the placenta.
You mention improved survival with the use of laser. This is true. I would highlight the interval from Eurofetus to the Solomon trial (2004-2015) as a good example of the improvements in survival over time. The Solomon trial is really your best example of contemporaneous survival statistics. The ones quoted are generous.
Figures 1 and 2 are very helpful. If I had my wish for this manuscript, I would provide a visual summary after each of the sections broken down by the donor and recipient. Again a wish and not a mandatory request. It would help process a large amount of information within this manuscript.
On the PDF there are multiple areas where the font changes.
Author Response
- Comment: In the abstract and introduction it is important to emphasize that you are highlighting the placental pathology and potential reasons for excess fetal loss. As a clinician who frequently treats TTTS with laser, I see the complications up close. Pre- and peri-viable delivery results in a significant proportion of the morbidity and mortality after laser as does placental insufficiency for the donor fetus as you do point out. Another area that is present is the significant cardiomyopathy seen in the recipient twins. So my point is the excess fetal loss despite laser comes from multiple sources rather than confined to just factors within the placenta.
Response: We agree with the reviewer’s suggestion and have added the sentence below.
Inserted sentences (Page 2, line 3-7): Clinically, fetal survival is difficult to predict because it can be affected by a combination of factors, including abnormal cord insertion, growth restriction due to differences in placental size, and preterm delivery due to postoperative rupture of the membrane [3], [11].
- Comment: You mention improved survival with the use of laser. This is true. I would highlight the interval from Eurofetus to the Solomon trial (2004-2015) as a good example of the improvements in survival over time. The Solomon trial is really your best example of contemporaneous survival statistics. The ones quoted are generous.
Response: We thank the reviewer for the useful comment. We have added new references on the solomon technique.
Inserted reference in the text: (Page 1, line 11 ): [5] [6] [7] [8] .
- Comment: Figures 1 and 2 are very helpful. If I had my wish for this manuscript, I would provide a visual summary after each of the sections broken down by the donor and recipient. Again a wish and not a mandatory request. It would help process a large amount of information within this manuscript.
Response: Accordingly, we have added a visual summary to the end of the “OS” section, because it was a bit complicated.
Inserted words (Page9, line23): (8-OHdG)
Inserted sentences (Page 9, line 46-47 ): ,because some OS-induced changes were observed in the placenta of the smaller twin of sFGR (Figure 3).
Inserted figure (Page 9, line 41): Figure 3
Inserted sentences in Figure 3 legend (Page 10, line 2-4): Figure 3. OS-induced changes in the placenta of smaller twin in sFGR. Abbreviations: HMOX1, heme oxygenase 1; MDA, malondialdehyde; mtDNA, mitochondrial DNA; NFE2L2, nuclear factor erythroid 2 like 2; 8-OHdG, 8-hydroxydeoxyguanosine
- Comment: On the PDF there are multiple areas where the font changes.
Response: We have changed the font of the main text and figures and figure legends accordingly.